# Comparative Molecular Characterization and Pharmacokinetics of IgG1-Fc and Engineered Fc Human Antibody Variants to Insulin-like Growth Factor 2 Receptor (IGF2R)

**DOI:** 10.3390/molecules28155839

**Published:** 2023-08-03

**Authors:** Chandra B. Prabaharan, Sabeena Giri, Kevin J. H. Allen, Katrina E. M. Bato, Therese R. Mercado, Mackenzie E. Malo, Jorge L. C. Carvalho, Ekaterina Dadachova, Maruti Uppalapati

**Affiliations:** 1Department of Pathology and Laboratory Medicine, College of Medicine, University of Saskatchewan, Saskatoon, SK S7N 5E5, Canada; chp347@usask.ca; 2College of Pharmacy and Nutrition, University of Saskatchewan, Saskatoon, SK S7N 5E5, Canada; byw294@usask.ca (S.G.); kja782@usask.ca (K.J.H.A.); mem510@mail.usask.ca (M.E.M.); auj201@mail.usask.ca (J.L.C.C.); 3Department of Anatomy, Physiology and Pharmacology, College of Medicine, University of Saskatchewan, Saskatoon, SK S7N 5E5, Canada; kmb027@mail.usask.ca (K.E.M.B.); trm545@mail.usask.ca (T.R.M.)

**Keywords:** osteosarcoma (OS), insulin-like growth factor 2 receptor (IGF2R), monoclonal antibodies, neonatal Fc receptor (FcRn), radioimmunotherapy (RIT)

## Abstract

Novel therapeutic approaches are much needed for the treatment of osteosarcoma. Targeted radionuclide therapy (TRT) and radioimmunotherapy (RIT) are promising approaches that deliver therapeutic radiation precisely to the tumor site. We have previously developed a fully human antibody, named IF3, that binds to insulin-like growth factor 2 receptor (IGF2R). IF3 was used in TRT to effectively inhibit tumor growth in osteosarcoma preclinical models. However, IF3’s relatively short half-life in mice raised the need for improvement. We generated an Fc-engineered version of IF3, termed IF3δ, with amino acid substitutions known to enhance antibody half-life in human serum. In this study, we confirmed the specific binding of IF3δ to IGF2R with nanomolar affinity, similar to wild-type IF3. Additionally, IF3δ demonstrated binding to human and mouse neonatal Fc receptors (FcRn), indicating the potential for FcRn-mediated endocytosis and recycling. Biodistribution studies in mice showed a higher accumulation of IF3δ in the spleen and bone than wild-type IF3, likely attributed to abnormal spleen expression of IGF2R in mice. Therefore, the pharmacokinetics data from mouse xenograft models may not precisely reflect their behavior in canine and human patients. However, the findings suggest both IF3 and IF3δ as promising options for the RIT of osteosarcoma.

## 1. Introduction

OS remains a formidable challenge in both human and canine patients, with the disease representing the most prevalent malignant primary bone tumor in both species [1,2]. OS has a dismal prognosis in canines, with pulmonary metastases developing in approximately 90% of cases and a median survival time of just four months for dogs treated with amputation alone [3,4]. While adjuvant carboplatin has improved median survival times (9–12 months) [5,6,7], alternative treatments like stereotactic radiation therapy [8,9,10] and tyrosine kinase inhibitors [11] have not yielded significant overall survival benefits for canine patients. Likewise, OS survival rates have remained largely stagnant in human patients at around 70% over the past 25 years [12,13,14], underscoring the pressing need for innovative therapeutic approaches to tackle this disease.

Targeted radionuclide therapy (TRT) is a more effective treatment option than external beam radiation therapy (EBRT). It delivers precise doses of radionuclides that emit cytocidal α or β particles to tumor cells [15,16] while minimizing damage to healthy tissues. TRT has been extensively researched in preclinical and clinical settings, leading to the approval of Xofigo^®^ (^223^Ra chloride) and Lutathera^®^ (^177^Lu-labeled peptide) for treating castration-resistant prostate cancer, bone metastases [17,18], and certain neuroendocrine tumors [19]. Pluvicto^®^ is a new addition to the TRT arsenal, a prostate-specific membrane antigen (PSMA)-binding molecule.

Radioimmunotherapy (RIT) is a targeted radiation therapy that uses therapeutic radionuclides attached to antibodies. This method effectively delivers therapeutic radionuclides throughout the body with fewer side effects [20]. Unlike other treatments, RIT does not rely on the immune system or tumor environment to be effective. The insulin-like growth factor 2 receptor (IGF2R) has been identified as an overexpressed target in all types of osteosarcoma (OS) cells [21], making it a suitable target for RIT. Studies have shown that mouse antibodies to IGF2R can significantly reduce OS xenograft growth without side effects [22,23], and IGF2R expression has been confirmed in canine OS [23]. 

IGF2R, also known as the cation-independent mannose 6-phosphate receptor (CI-M6PR), is a membrane protein highly conserved in vertebrates and expressed ubiquitously in normal tissues [24]. It interacts with ligands such as mannose 6-phosphate containing glycoproteins and Insulin-like growth factor 2 (IGF-II). It mainly directs these ligands to the lysosomal degradation pathway to regulate protein activity [25]. Recently, IGF2R has been utilized as a target for the potential degradation of oncoproteins using Lysosome-Targeting Chimaeras (LYTAC) [26]. The role of IGF2R as an oncogene is unclear, with contradicting reports suggesting that it has tumor-suppression activity in certain tumors [27]. Nevertheless, for effective RIT, the high expression of IGF2R in OS is sufficient to deliver cytocidal doses of radiation to the tumor, while sparing normal tissues.

Human antibodies to IGF2R have been developed, allowing for further evaluation of RIT efficacy and safety in mice and dogs with human and canine OS xenografts, respectively [28]. A human IgG antibody called IF3 was generated using phage display technology. It was designed to bind specifically to domains 11–14 of IGF2R that contains the IGF-II binding site and binds to conserved epitopes across human, mouse, and canine IGF2R [29]. ^177^Lu-labelled IF3 antibody demonstrated efficacy in reducing tumor growth in a canine osteosarcoma xenograft model. Nonetheless, the antibody’s half-life was limited to less than 4 h, and there was a significant buildup of radiolabeled IF3 in the spleen of SCID mice [28]. The abnormal uptake of the spleen was found to be mouse-specific. There was minimal spleen uptake of ^89^Zr-labeled IF3 in dogs during PET/CT imaging, and tissue cross-reactivity studies in canine spleen and in human major tissues showed only the intracellular location of IGF2R, which precludes antibody binding to healthy organs [29,30]. 

Given the role of IGF2R in the lysosomal degradation of ligands and ubiquitous low-level expression in normal tissues, target mediated drug-disposition (TMDD) [25] appears to be the primary pathway of the degradation of IF3. The neonatal FcRn-receptor-based endosomal recycling pathway can redirect IF3 away from TMDD pathway and potentially improve the circulation levels of IGF2R antibodies. The Fc region of immunoglobulins binds to the FcRn receptor in cells found in almost all organs in mammals, avoiding degradation with endolysosomal pathways [31,32,33]. This binding occurs at endosomal pH (5.5–6.0), and the IgG–FcRn complex is directed back to the plasma membrane instead of the endolysosomal compartment [31,34]. Upon fusion with the plasma membrane, the complex is released back into circulation due to the low affinity of FcRn for the Fc domain at extracellular pH 7.4. Several groups have optimized the binding interface of the IgG—FcRn complex to improve binding affinity at pH 5.8 for retention and reduced binding affinity at pH 7.4 for effective release to improve the half-life of antibodies [31,35]. 

In this study, we used a combination of L309D Q311H and N434S (termed DHS mutation), which has been shown to increase the half-life of IgGs in humans to 72 days [31]. We hypothesized that this variant of IF3 (termed IF3δ) will have improved pharmacokinetics in mice compared to the wildtype IF3. We expressed both naïve (IF3) and Fc-engineered versions (IF3δ), characterized in vitro binding, and compared the biodistribution of both ^111^In-labeled antibodies in SCID and C57BL/6 mice.

## 2. Results

### 2.1. Expression, Purification, and Characterization of IF3 and IF3δ

To enhance the pharmacokinetic (PK) properties of the wildtype IF3, a modified version of an IgG1, a monoclonal antibody (mAb) known as IF3δ was developed. This involved incorporating three specific amino acid substitutions (L309D, Q311H, and N434S) within the Fc fragment of the mAb based on published studies [31].

We expressed and purified IF3 by transient transfection in Expi293F cells as described in methods [36]. For expressing IF3δ, we used a bicistronic pTRIOZ-hIgG1 vector (Invivogen, San Diego, CA, USA). We modified the vector to replace the Fc region of the heavy chain with a gene fragment containing corresponding DHS mutations (Appendix A). The antibodies were purified using protein A column, an affinity chromatography technique. The IgG versions of IF3 and IF3δ expression levels were 10 mg/L. The purified proteins were analyzed using SDS-PAGE (Appendix A). Further, IF3 and IF3δ antibodies were conjugated using the bifunctional chelator CHXA″-DTPA. Both CHXA″-DTPA conjugated and unconjugated versions of IF3 and IF3δ were subjected to ELISA and showed specific bindings to human, canine, and mouse IGF2R with a nanomolar affinity (Figure 1) (Table 1). The specific binding was also analyzed with flow cytometry (Figure 2), and we found that IF3δ (DHS mutation) showed binding similar to wildtype IF3. Additionally, it was observed that there was no loss of affinity due to the Fc mutation.

We utilized a pH-sensitive dye to investigate the internalization of IF3 and IF3 (IgG1) in Gracie patient-derived OS cells. The dye remains non-fluorescent under neutral pH but turns highly fluorescent under acidic conditions. We labeled the antibody with the dye to track its internalization process. The antibody did not exhibit fluorescence upon binding to its antigen on cancer cells. However, it underwent internalization into endosomes and lysosomes, causing the pH to drop and leading to fluorescence activation of the dye. This approach allowed us to examine antibody internalization, which is crucial in radionuclide retention within tumors, while also contributing to TMDD. The results of the antibody internalization experiment were conclusive; within 48 h of introduction to OS, IF3 and IF3δ mAb were significantly internalized. Conversely, the non-specific control RSV mAb demonstrated minimal internalization (Figure 3). 

### 2.2. pH 5.8 Binding of IF3 and IF3δ Antibodies to Human and Mouse FcRn Receptors

We investigated the binding of the IF3 and IF3δ (DHS mutation) antibodies to recombinant human and mouse FcRn receptors (R&D systems), specifically at pH 5.8. Using ELISAs, we measured the binding relative to wild-type IF3 and determined the IC_50_ values (Figure 4). Our results demonstrate strong IF3 and IF3δ antibodies binding to FcRn receptors at pH 5.8. Notably, the overall binding affinity between wild-type IF3 and IF3δ was comparable between human and mouse versions, indicating that the DHS mutation did not significantly affect the affinity towards a mouse version of FcRn, which can lead to improved half-life in mouse serum.

### 2.3. Biodistribution Comparison of ^111^In-Radiolabeled IF3 and IF3 Antibodies in SCID and C57BL/6 Mice Reveals Differences in Accumulation in Spleen and Bone

Biodistribution studies were conducted in SCID and C57BL/6 black mice using ^111^In-radiolabeled IF3 and IF3δ antibodies (Figure 5). The study results showed that both radiolabeled IF3 and IF3δ antibodies exhibited a similar biodistribution pattern in most organs, including the liver, kidney, and lung, in both SCID and C57BL/6 mice. Both antibodies were found to have a rapid clearance from the bloodstream, with only approximately 2% of the injected dose per gram (ID/g) remaining after 24 h post-injection.

However, significant differences were observed in the accumulation of these antibodies in the spleen and bone. In the case of SCID mice, the IF3δ antibody exhibited a lower accumulation in the spleen (58.72%) compared to the IF3 antibody (68.70%) at 4 h post-administration. The accumulation of IF3δ antibodies in the spleen increased further at 24 h post-administration (106.92%), whereas the IF3 antibody accumulation remained relatively constant (51.62%). In the case of C57BL/6 black mice, the IF3δ antibody showed a significantly higher accumulation in the spleen (117.39%) compared to the IF3 antibody (48.79%) at 4 h post-administration. At 24 h post-administration, the accumulation of IF3δ in the spleen remained higher than the IF3 (103.31% vs. 51.62%).

Similarly, at 4 h post-administration, the radiolabeled IF3δ showed greater bone presence than IF3 in both SCID and C57BL/6 mice. The same was observed in blood, where the presence of IF3δ was lower than IF3 in both mouse types. After 24 h, the IF3 delta antibody presence in bone decreased, yet remained higher than the IF3 antibody in both mouse types. The previously observed differences in blood between the two types of antibodies were no longer present.

Furthermore, the variations noticed between the IF3 and IF3δ antibodies were influenced by the specific strain of the mouse used.

## 3. Discussion

As part of our efforts to optimize the characteristics of the IF3 antibody for preclinical and clinical applications, we generated a mutant antibody known as IF3δ. This antibody was designed to have improved pharmacokinetic properties, such as slower blood clearance, compared to the wild-type IF3 antibody. Our previous research showed that the original IF3 antibody was rapidly cleared from the blood of SCID mice with osteosarcoma xenografted tumors [28,29]. Additionally, these studies showed that targeting IGF2R on canine-patient-derived osteosarcoma tumors in mice with radioimmunotherapy resulted in effective tumor regression [28]. However, the use of the IF3 antibody for this purpose was limited due to its rapid clearance and accumulation in the spleen, resulting in a spleen sink [28,29]. Therefore, to address this issue and improve the efficacy of the IF3 antibody for potential clinical use, we generated a Fc-engineered antibody, IF3δ. The IF3δ antibody was expected to have a longer circulation time in the blood [31].

Our ELISA experiment shows that DHS mutation did not significantly affect the binding to mouse FcRn at pH 5.8 and retained tight binding to human, mouse, and canine IGF2R. Therefore, we expected to see an increase in IF3δ levels in the blood, which can lead to better tumor targeting and effectivity.

Biodistribution studies conducted in SCID and C57bl/6 mice revealed that radiolabeled monoclonal antibodies IF3 and IF3δ primarily accumulate in the spleen, with the DHS mutation resulting in greater splenic accumulation than the wild-type IF3. We used female SCID and female immunocompetent mice to compare the biodistribution of IF3 and IF3-delta. We have chosen female SCID mice, as this was the model that we previously utilized in tumor uptake and therapy studies [28,29], and sex-matching healthy mice were supposed to have less spleen uptake. Nevertheless, these animal models had significant limitations due to the abnormal binding of human monoclonal antibodies to the murine Fc receptors in the spleen and bone marrow [37]. In future studies, we intend to use humanized mice of both sexes with human Fc receptors [38] to overcome the problem of this abnormal binding. In addition, during the comparative oncology stage of the IF3-delta development project, we will use healthy beagle dogs of both sexes, as we have completed in the recent IF3 imaging and dosimetry study [30]. Finally, as human antibodies are known to have prolonged (up to several weeks) half-lives in human patients [34,35], we anticipate that the effective half-life of IF3-delta in humans will give it sufficient time to achieve a high tumor uptake but not exert toxicity to the bone marrow and other healthy tissues.

Several factors influence the disposition of the antibodies (IF3 and IF3δ). Firstly, the antibody interacts with the FcRn receptor for recycling while competing with Fcγ receptors in splenic macrophages, monocytes, and neutrophils to be internalized and eliminated [39,40,41]. Secondly, the expression of the target IGF2R in the mouse spleen affects the disposition of the antibody [23]. Additionally, IGF2R acts as a lysosome-targeting receptor (LTR) and can efficiently deliver proteins bearing N-glycans capped with mannose 6-phosphate (M6P) or other ligands to lysosomes [26]. Our internalization studies on OS cells derived from a canine patient, Gracie, support the statement. Finally, the clearance rate is influenced by the mouse strain, with SCID and NSG mice known to have anomalous distribution patterns of human IgG1 [41,42]. The relatively short circulation half-life of monoclonal antibodies IF3 and IF3δ in the bloodstream can be attributed to the phenomenon of “spleen sink”, whereby a significant portion of the administered antibody dose accumulates in the spleen. Thus, to address these limitations, potential strategies such as co-administration of unlabeled antibodies and strategies to block Fcγ receptors—such as HVIG or other Fc blockers—can be used in future subsequent studies [41]. Importantly, the study conducted in healthy beagle dogs to estimate the biodistribution of ^89^Zr-IF3 using image-based dosimetry demonstrated that, unlike in mouse models, there was no detectable deposition in normal bone, and IF3 did not significantly accumulate in the spleen [43]. This suggests that pharmacokinetic data from preclinical studies in mice might not reflect the IF3 distribution of canine and human patients. It is possible that the biodistribution of IF3δ can be significantly better than that of IF3 in OS patients, and further studies in dogs need to be conducted. Further research is necessary to investigate the influence of glycosylation, charge, and isoelectric point [37] on IF3 antibody pharmacokinetics.

## 4. Materials and Methods

### 4.1. Construction of Mammalian Expression Vector with DHS Mutation

The pTRIOZ IgG1(bicistronic) vector (Invivogen, San Diego, CA, USA) containing both VL and VH cassette was utilized to introduce the DHS mutation in the Fc region, involving L309D Q311H and N434S substitutions (Appendix A) [31]. A gene fragment containing the DHS mutation was obtained from Integrated DNA Technologies (Coralville, IA, USA). The vector was digested with MluI and AvrII and the gene fragment was ligated using Gibson Assembly. The resulting ligated vector was then transformed into NEB 5α Escherichia coli strain (NewEngland Biolabs, Whitby, ON, Canada). Subsequently, colonies were selected and sequenced to confirm the presence of the DHS mutation within the Fc region.

### 4.2. Production of IgG1 Antibodies

To produce IgG lead antibodies, the coding sequences for VL and VH of IF3 were inserted into pFUSE2ss-CLIg-hK, pFUSE2ss-CHIg-hG1, and pTRIOZ-hIgG1 vectors containing DHS mutation, respectively. These constructs were then expressed in Expi293F cells using established methods [36]. To express IF3, a two-vector system was used with pFUSE2ss-CLIg-hK and pFUSE2ss-CHIg-hG1. For IF3δ, a single vector system with pTRIOZ-hIgG1 was used. The resulting IF3 and IF3δ antibodies were purified using MabSelectSure affinity resin as recommended by the manufacturer. SDS-PAGE analysis assessed the proteins’ purity (Appendix A).

### 4.3. Cell Lines 

The patient-derived osteosarcoma cell line OS33 was provided by Dr. R. Gorlick at MD Anderson Cancer Center in Houston, TX. The canine-patient-derived osteosarcoma cell line Gracie was provided by Dr. Doug Thamm’s laboratory at Colorado State University School of Veterinary Medicine. The OS33 cells were cultured in Dulbecco’s Modified Eagle Medium, while Gracie cells were cultured in RPMI-1640 with HEPES. All media were supplemented with 10% fetal bovine serum, sodium pyruvate, non-essential amino acids, and 100 U penicillin/0.1 mg/mL streptomycin.

### 4.4. ELISA for Purified IgG1 Antibodies against IGF2R

The recombinant human, mouse, and canine IGF2R were coated on a NUNC Maxisorp 96-well plate at 3 μg/mL concentration in PBS, with 50 μL (0.15 μg) added to each well. After being blocked with PB Buffer (1 × PBS + 0.2 mg/mL BSA) for an hour, the plate was washed twice with PT Buffer (1 × PBS + 0.05% Tween20). Purified IgGs (IF3 and IF3δ) were then added to the corresponding wells in concentrations ranging from 0.02 nM to 500 nM (1:3 dilution), and Palivizumab (IgG1) against the respiratory syncytial virus (RSV) was used as a negative control and incubated for 1 h with shaking. After washing four times with PT buffer, the plate was treated with a secondary antibody, Goat Anti-Human Kappa (K) Light Chain, HRP (Invitrogen, Waltham, MA, USA), diluted 1:5000 in PBT buffer. The plate was incubated for 45 min with shaking, then washed four times with PT buffer and developed with TMB 5,5′-Tetramethylbenzidine substrate for approximately 5 min. To terminate the reaction, 45 μL of 1 M Phosphoric acid was added to each well and read at 450 nm using a Biotek Synergy microplate reader.

### 4.5. ELISA for Purified IgG1 Antibodies against FcRn

The recombinant human and mouse FcRn (R&D systems 8639-FC-050 and 9114-FC-050) were coated on NUNC Maxisorp 96-well plate at a working concentration of 2 ug/mL added to each well. All the buffers and reagents for the experiments were made at pH 5.8. The plates were blocked using PB buffer (1 × PBS + 0.2 mg/mL BSA) for an hour, and the plates were washed twice with PT buffer (1 × PBS + 0.05% Tween20). IF3 and IF3δ were then added to the corresponding wells in a concentration ranging from 1000 nM–0.02 nM (1:3 dilution) at pH 5.8 and incubated for 1 h in an orbital shaker. After being washed four times with PT buffer, the plates were treated with a secondary antibody, Goat Anti-Human Kappa (K) Light Chain, HRP (Invitrogen), diluted 1:5000 in PBT buffer. The plate was incubated for 45 min with shaking, then washed four times with PT buffer and developed with TMB substrate for approximately 5 min. To terminate the reaction, 45 μL of 1 M Phosphoric acid was added to each well and read at 450 nm using a Biotek Synergy microplate reader.

### 4.6. Flow Cytometry

In each well, 3 × 10^5^ cells were placed in a 96-well plate, and different concentrations of primary antibodies (IF3, IF3δ) ranging from 25 nM to 220 nM were added and incubated for 30 min. Negative control was also included using an anti-RSV antibody (palivizumab). After incubation, the cells were washed twice with FACS buffer, and a secondary antibody was added (Goat Anti-Human IgG Fc (PE) for IF3 and IF3δ) and incubated for another 30 min. The cells were washed three times with FACS buffer before being analyzed using a CytoFlex machine.

### 4.7. Antibody Internalization Assay

Cells of the Gracie OS type were seeded in a microplate with 96 wells, with a concentration of 4000–5000 cells per well in 50 μL of RPMI medium. After 24 h, the IncuCyte FabFluor Red Antibody Labeling Reagent was rehydrated using 100 μL of sterile water (final concentration: 0.5 mg/mL). The IF3, IF3δ, and RSV (Palivizumab) IgG1-negative control antibodies were mixed with the labeling reagent in a 1:3 molar ratio and then incubated at 37 °C for 15 min. The antibody–reagent mixture was added to the cells at a concentration of 4 μg/mL. The IncuCyte Live-cell Analysis System with a 10× objective captured images every 15 min for 48 h. The red fluorescent channel and software features were used to subtract the background fluorescence.

### 4.8. Animal Models 

Severe combined immunodeficient female mice (SCID CB17/Icr-Prkdcscid/IcrIcoCrl) and C57BL/6 immunocompetent female mice were obtained from Charles River Laboratories in Wilmington, MA. Biodistribution experiments were carried out in 20 SCID (5 mice/group) and 20 C57BL/6 (5 mice/group) using ^111^In-radiolabeled IF3 and IF3δ antibodies. 

### 4.9. Conjugation and Radiolabeling of Antibodies

IF3 and IF3δ antibodies were conjugated with the bifunctional chelating agent CHXA″-DTPA (*R*)-2-Amino-3-(4-isothiocyanatophenyl)propyl]-trans-(S,S)-cyclohexane-1,2-diamine-pentaacetic acid) using a 2.5-fold molar excess, and the conjugated antibodies were purified using Amicon concentrators with 0.15 M ammonium acetate buffer (pH 6.5–7.0). The conjugated antibodies were stored at 4 °C until further use. The conjugate–antibody ratio (CAR) of CHXA″-DTPA molecules per IF3 or IF3δ molecule post-conjugation was determined on a Bruker Ultraflextreme MALDI-TOF (Bruker, Bremen, GmbH) to be 0.91 and 0.84. The conjugated antibodies were radiolabeled with ^111^In for biodistribution at a specific activity of 10:1 μCi/ug using ^111^In for IF3. The reaction mixture was incubated for 45 min at 37 °C with shaking and then free metals are quenched using 3 μL of 0.05 M EDTA solution. The radiolabeling percentage was measured using SG-iTLC strips with 0.15 M ammonium acetate buffer as the solvent phase. Radiolabeling yields were greater than 95% for IF3 and 92% for IF3δ. To achieve 99% radiochemical purity, the radiolabeled antibodies were purified using size-exclusion spin filters. The radiochemical purity of the radiolabeled antibody was analyzed using high-performance liquid chromatography (HPLC) to ensure the quality of the final product (Appendix A).

### 4.10. Biodistribution 

SCID and C57BlL/6 mice were randomized into four groups (5 mice/group) and injected intravenously with 30 µCi of ^111^In-IF3 and ^111^In-IF3δ. At 4 h and 24 h post-injection, mice were sacrificed, and the following organs were collected: blood, heart, lungs, pancreas, spleen, kidney, liver, brain, stomach, small intestine, large intestine, thigh muscle, and bone. The percentage injected dose per gram (% ID/g) was then calculated by weighing the organ and counting the radioactivity with a gamma counter (PerkinElmer, Waltham, MA, USA).

### 4.11. Statistical Analysis

Biodistribution differences were analyzed among groups utilizing the Kruskal–Wallis and/or the Mann–Whitney tests for non-normally distributed data. A two-way ANOVA test was implemented to compare the impacts of time (4 h and 24 h) and treatment (IF3 and IF3δ) on the treatment groups. Post hoc comparisons between the 4 h and 24 h groups (IF3 and IF3δ) were conducted using Tukey’s Honestly Significant Difference (HSD) test to determine specific biodistribution differences in GraphPad prism software 8.0. These statistical analyses ensured the accuracy and validity of the results for the biodistribution study, allowing us to identify significant differences between experimental groups and evaluate the impact of time and treatment on biodistribution patterns. 

## 5. Conclusions

This study aimed to improve the pharmacokinetic properties of the IF3 antibody, which targets IGF2R, by introducing a DHS mutation in the Fc region. The IF3δ mutant antibody demonstrated a similar binding affinity to its target receptors as the wild-type IF3 antibody. Biodistribution experiments in mice showed increased spleen accumulation with IF3δ compared to IF3. However, different mouse strains displayed variations in bone accumulation. The DHS mutation improved certain aspects of antibody performance, but further optimization is needed to maximize its therapeutic efficacy in osteosarcoma treatment.

## Figures and Tables

**Figure 1 molecules-28-05839-f001:**
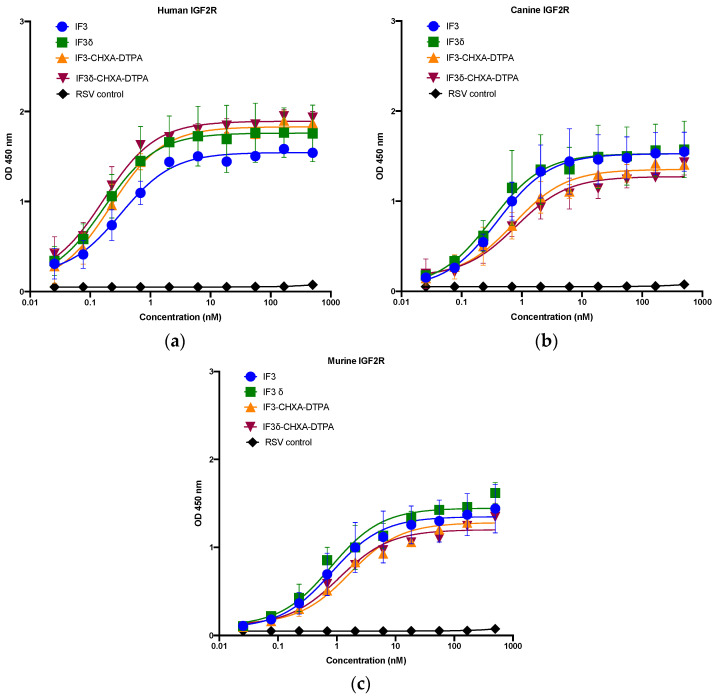
Binding ELISA of unconjugated IgGs IF3 and IF3δ and CHXA″-DTPA-conjugated antibodies to recombinant (**a**) human, (**b**) canine, and (**c**) mouse IGF2R–Fc proteins. Palivizumab (IgG1) against the respiratory syncytial virus (RSV) was used as a negative control. (EC_50_ values for IF3 and IF3δ are indicated below in Table 1, Mean ± SEM, n = 3).

**Figure 2 molecules-28-05839-f002:**
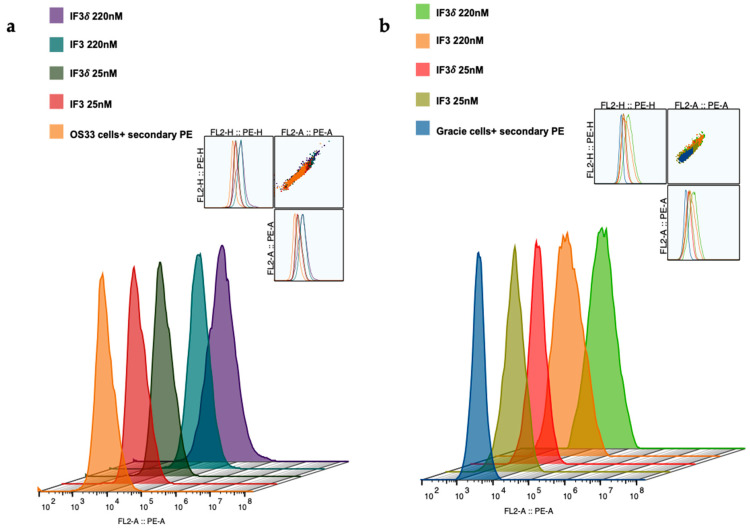
Binding of IF3 and IF3δ mAbs at different concentrations to human- and canine patient-derived OS cells using flow cytometry. (**a**) OS-33 (human); (**b**) Gracie (canine). Goat Anti-Human IgG PE was used as a secondary antibody. (The insert represents cell population).

**Figure 3 molecules-28-05839-f003:**
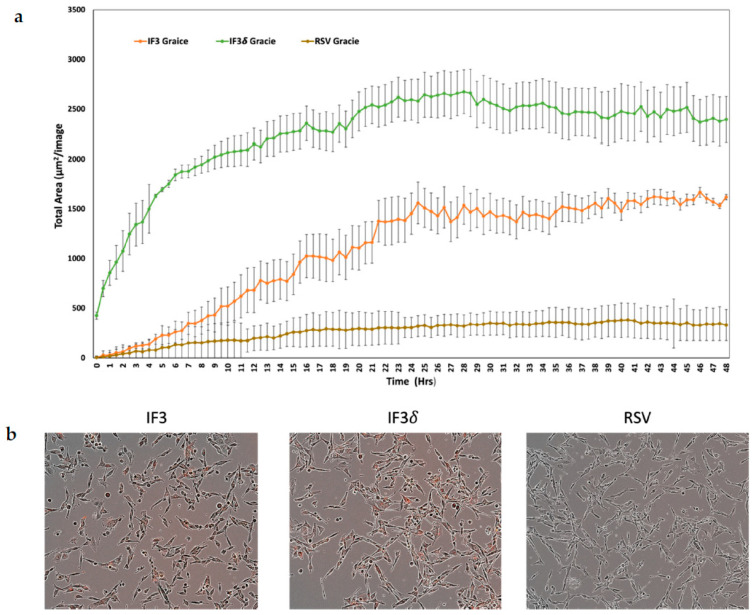
Receptor-mediated internalization of anti-IGF2R antibodies; IF3, IF3δ, and Palivizumab (IgG1) against the respiratory syncytial virus (RSV) used as a negative control were conjugated to IncuCyte FabFluor Red Antibody Labeling Reagent, and internalization into Gracie OS cells was analyzed at 48 h post-binding. (**a**) The graph shows the total of red areas in each image captured using the IncuCyte system every 30 min for 48 h (three independent experiments). The standard deviation is represented by the error bars on the plot. (**b**) The left insert displays Gracie cells incubated with IF3 mAb, the middle insert shows cells treated with IF3δ, and the insert on the right displays cells exposed to an irrelevant RSV mAb.

**Figure 4 molecules-28-05839-f004:**
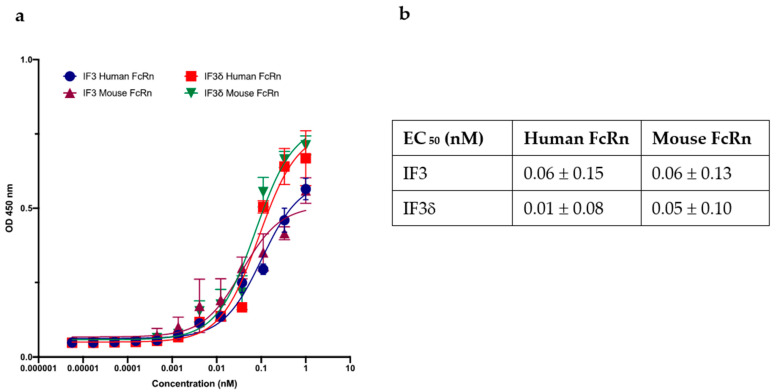
The effect of wild-type and DHS-mutated IgG1 on binding to FcRn at endosomal acidic pH 5.8. (**a**) IgG IF3 and IFδ binding to recombinant human neonatal Fc receptor (FcRn) at pH 5.8. (**b**) EC_50_ values (Mean ± SEM, n = 3) for IF3 and IF3δ are indicated above.

**Figure 5 molecules-28-05839-f005:**
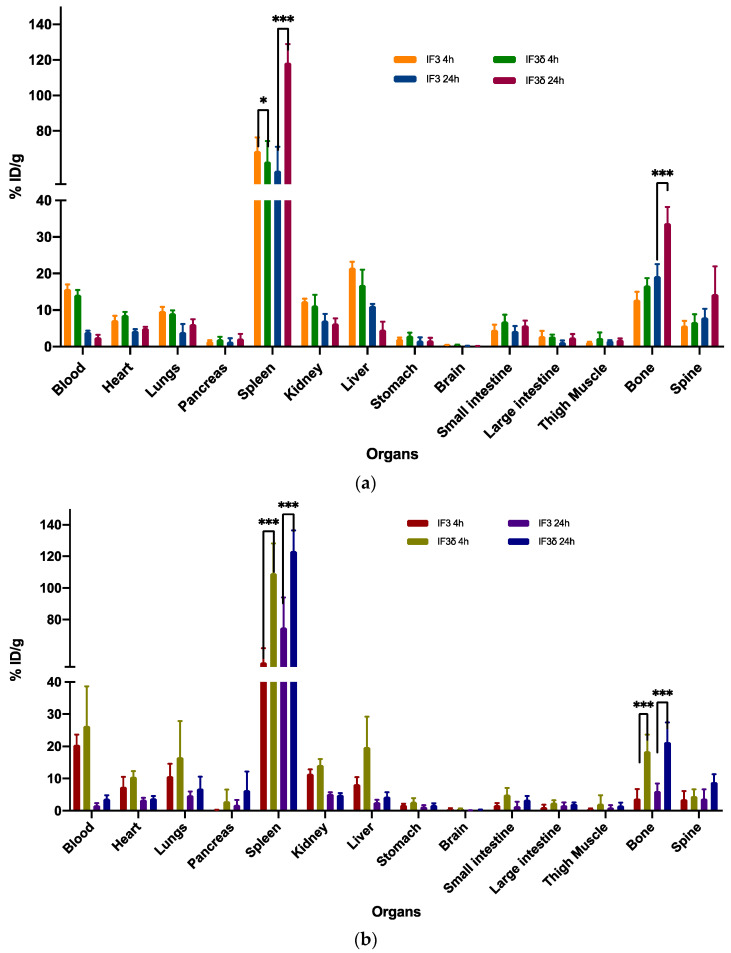
The biodistribution of ^111^In-labeled IF3 and IF3 in female mice (**a**) SCID and (**b**) C57BL/6 was measured at 4 and 24 h post-antibodies-administration. The results showed that the uptake of IF3 mAb was significantly higher than IF3 in the spleen for both mouse strains at both time points (*p*-value; *** =< 0.0001; * = 0.03). Additionally, the bone uptake was significantly higher in C57BL/6 at both time points, while in SCID mice, the difference in bone uptake was only observed at the 24 h timepoint (*p*-value; *** =< 0.0001). The Y-axis represents the % of injected dose per gram (% ID/g); Data are mean ± SD, n = 5 mice/study group.

**Table 1 molecules-28-05839-t001:** EC_50_ values of IF3 and IF3δ antibodies for human, mouse, and canine IGF2R.

EC _50_ (nM)	Human IGF2R	Canine IGF2R	Mouse IGF2R
IF3	0.22 ± 0.45	0.41 ± 0.75	0.81 ± 1.77
IF3δ	0.17 ± 0.41	0.32 ± 0.75	0.79 ± 1.46
CHXA″-DTPA-IF3	0.15 ± 0.32	0.73 ± 1.28	1.65 ± 2.63
CHXA″-DTPA-IF3δ	0.11 ± 0.24	0.79 ± 1.49	1.1 ± 1.64

## Data Availability

The article and Appendix A contains all the data pertaining to this study.

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
