# Peer review of "Comparative Molecular Characterization and Pharmacokinetics of IgG1-Fc and Engineered Fc Human Antibody Variants to Insulin-like Growth Factor 2 Receptor (IGF2R)"

_molecules, 2023, doi:10.3390/molecules28155839_

Round 1

Reviewer 1 Report

Dear Authors,

The paper titled "Comparative molecular characterization and pharmacokinetics 2 of IgG1-Fc and engineered Fc human antibody variants to insu-3 lin growth factor receptor type 2 (IGF2R)" was well written. I appreciate your effort to conduct the study and you tried to improve the half-life of IF3 in the present article from your previous publication. The study has been well executed and appropriate models are used and the methodology is explained in a neat manner. Hope the addition of the canine model would have added more value to the current study as the biodistribution of IF3delta in the spleen would have been reduced.

Author Response

We appreciate the reviewer’s comments supporting the manuscript.

In response to “Hope the addition of the canine model would have added more value to the current study as the biodistribution of IF3delta in the spleen would have been reduced.”

We have recently published the IF3 uptake and dosimetry studies in healthy dogs in a separate manuscript (https://doi.org/10.3390/ph16070979). We hope to be able to perform a similar study for IF3delta.

Reviewer 2 Report

In this manuscript, the authors develop an antibody named IF3δ with the aim of enhancing binding affinity and biodistribution compared to its precursor form IF3. According to the experimental results, the newly created antibody demonstrates a similar binding affinity to the wild-type IF3 but shows variable accumulation in certain tissues, particularly in the spleen and bone. The authors employ a straightforward approach to compare the binding potency of the mutated IF3 and its biodistribution in various tissues with that of the wild-type IF3. However, a crucial concern arises from the absence of statistical tests in the figures, which undermines the scientific credibility of the results and conclusions. To ensure the reliability of their findings, the authors should carefully examine the data and include appropriate statistical significance levels when drawing conclusions, especially in the comparative analysis.

Author Response

We thank the reviewer for the comments to improve this manuscript.

“However, a crucial concern arises from the absence of statistical tests in the figures, which undermines the scientific credibility of the results and conclusions. To ensure the reliability of their findings, the authors should carefully examine the data and include appropriate statistical significance levels when drawing conclusions, especially in the comparative analysis.”

We have added a section on statistical analysis methodology for the comparative biodistribution study.  The p-values showing statistical significance are added in Figure 5. The sample size and standard error was mentioned for data in Figures 1, 3 and 4.

Reviewer 3 Report

There are some issues with the format. Such as The abstract should be shorter, less than 250 words. There is a space between the number and symbol, such as 0.2mg/ml should be 2.2 mg/mL. and more examples could be found in Table 1 and many more other places. The cited unpublished data should be a reference should be added in the References section. It is the author's responsibility to check the manuscript correct format before submission. The author should revise the format of the manuscript.

In addition, some detail information for the experiment, such as the sample size, the control, etc is missing. Due to the missing information of the design of the experiment,  it is hard to make a decision that this is statistical significant. 

A figure caption should be standalone. A short legend should be added to each figure including methods so the reader can understand the figures without reading the texts. And the sample size of the experiment should add. The statistical information such as statistical tests,  p valve should add to the figure. 

Why the author choose EC50 not IC50? And did the author check the cytotoxicity? Is there a reason only female mice?

The author need to mention the meaning of ID in figure 5.

A figure or table about the antibody and/or pathway and/or working model would draw more reader's interests. 

L275 E coli should be (Escherichia coli)?

Add full name for the abbreviation, such as IGF-II, LYTAC, IF3 etc. 

A bit discussion about the animal model should be added. 

English is good. 

Author Response

We thank the reviewer for the detailed comments to improve the manuscript.

“There are some issues with the format. Such as The abstract should be shorter, less than 250 words. There is a space between the number and symbol, such as 0.2mg/ml should be 2.2 mg/mL. and more examples could be found in Table 1 and many more other places. The cited unpublished data should be a reference should be added in the References section. It is the author's responsibility to check the manuscript correct format before submission. The author should revise the format of the manuscript.”

We updated the abstract to less than 200 words (as per the author guidelines). We carefully went through the manuscript and inserted space between number and symbol/units. The cited data is now published at https://doi.org/10.3390/ph16070979 and we have included the new reference.

“In addition, some detail information for the experiment, such as the sample size, the control, etc is missing. Due to the missing information of the design of the experiment,  it is hard to make a decision that this is statistical significant.

A figure caption should be standalone. A short legend should be added to each figure including methods so the reader can understand the figures without reading the texts. And the sample size of the experiment should add. The statistical information such as statistical tests,  p valve should add to the figure.”

We have added a section on statistical analysis methodology for the comparative biodistribution study.  The p-values showing statistical significance are added in Figure 5. The sample size and standard error was mentioned for data in Figures 1, 3 and 4. Control data was previously present in the Figures and is now highlighted in the Figure Legends. We have added more description to the figure legends and p-values were added in Figure 5.

“Why the author choose EC50 not IC50? And did the author check the cytotoxicity? Is there a reason only female mice?”

Our objective here was to compare the differences in binding and felt that EC50 using a direct binding ELISA will enable us to determine such differences.

We did not check the cytotoxicity, due to the abnormal biodistribution in mice. We have conducted safety studies in three canine patients, and they showed good tolerance to injected doses of radiolabeled IF3.We did not consider the sex-based differences in this preliminary study. We note the reviewer’s concern and we have included male and female dogs in the recently published canine study https://doi.org/10.3390/ph16070979.

“The author need to mention the meaning of ID in figure 5.”

Added to the figure legend.

“A figure or table about the antibody and/or pathway and/or working model would draw more reader's interests.”

Radioimmunotherapy does not rely on agonistic or antagonistic activity of the antibody and mediates cytotoxicity by DNA damage and apoptosis. We are not sure how a figure of mechanism will enhance the manuscript and we believe that it is not necessary.

“L275 E coli should be (Escherichia coli)?”

Corrected

“Add full name for the abbreviation, such as IGF-II, LYTAC, IF3 etc.”

We have added full descriptions for abbreviations where needed. IF3 is a name not an abbreviation.

“A bit discussion about the animal model should be added.”

It is not clear where the discussion of the animal models was lacking and wish that the reviewer elaborated more on this. We believe that we have described/provided references for the mouse xenograft models and discussed the anomalous behavior in the Discussion section.

Round 2

Reviewer 3 Report

the full name of injected dose per gram ID/g should be in Line 189, not L373

L357, can the author add the manufacture of the MALDI-TOF

add the sample size to 4.8 Animal models and figure 5 as well

“A bit discussion about the animal model should be added.” Such as the author the chosen of only female, and is there any other research using this model, the limitation of this mouse model compared to dog model, extending a bit about the idea of It is possible that the biodistribution of IF3 can be 259 significantly better than that of IF3 in OS patients. Of course, the discussion is already a good shape but it would be nice to extend a bit. 

Author Response

We thank the reviewer for the additional comments. Our responses are noted below:

"the full name of injected dose per gram ID/g should be in Line 189, not L373"

Added in Line 192

"L357, can the author add the manufacture of the MALDI-TOF"

Added in Line 377

"add the sample size to 4.8 Animal models and figure 5 as well"

Added in Lines 366-367 (section 4.8) and 223 (Figure 5)

"“A bit discussion about the animal model should be added.” Such as the author the chosen of only female, and is there any other research using this model, the limitation of this mouse model compared to dog model, extending a bit about the idea of It is possible that the biodistribution of IF3 can be 259 significantly better than that of IF3 in OS patients. Of course, the discussion is already a good shape but it would be nice to extend a bit."

A paragraph was added in Lines 245-258 to address this.